# Risk factors associated with SARS-CoV-2 infection in a multiethnic cohort of United Kingdom healthcare workers (UK-REACH): A cross-sectional analysis

Christopher A. Martin[1,2], Daniel Pan[1,2], Carl Melbourne[3], Lucy Teece[4], Avinash Aujayeb[5], Rebecca F. Baggaley[1], Luke Bryant[1], Sue Carr[6,7], Bindu Gregary[8], Amit Gupta[9], Anna L. Guyatt[10], Catherine John[10], I Chris McManus[11], Joshua Nazareth[1,2], Laura B. Nellums[12], Rubina Reza[13], Sandra Simpson[14], Martin D. Tobin[10], Katherine Woolf[11], Stephen Zingwe[15], Kamlesh Khunti[16], Keith R. Abrams[17], Laura J. Gray[4], Manish Pareek[1,2]*, UK-REACH Study Collaborative Group[¶]

1 Department of Respiratory Sciences, University of Leicester, Leicester, United Kingdom, 2 Department of Infection and HIV Medicine, University Hospitals of Leicester NHS Trust, Leicester, United Kingdom, 3 Genetic Epidemiology Research Group, Department of Health Sciences, University of Leicester, Leicester, United Kingdom, 4 Biostatistics Research Group, Department of Health Sciences, University of Leicester, Leicester, United Kingdom, 5 Respiratory Department, Northumbria Specialist Emergency Care Hospital, United Kingdom, 6 Department of Nephrology, University Hospitals of Leicester NHS Trust, Leicester, United Kingdom, 7 General Medical Council, London, United Kingdom, 8 Lancashire Clinical Research Facility, Royal Preston Hospital, United Kingdom, 9 Oxford University Hospitals NHS Foundation Trust, Oxford, United Kingdom, 10 Department of Health Sciences, University of Leicester, Leicester, United Kingdom, 11 University College London Medical School, London, United Kingdom, 12 Population and Lifespan Sciences, School of Medicine, University of Nottingham, Nottingham, United Kingdom, 13 Centre for Research & Development, Derbyshire Healthcare NHS Foundation Trust, Derby, United Kingdom, 14 Nottinghamshire Healthcare NHS Foundation Trust, Nottingham, United Kingdom, 15 Research and Development Department, Berkshire Healthcare NHS Foundation Trust, Bracknell, United Kingdom, 16 Diabetes Research Centre, University of Leicester, Leicester, United Kingdom, 17 Department of Statistics, University of Warwick, United Kingdom

[¶]Membership of UK-REACH Study Collaborative Group is provided in the Acknowledgments.
* manish.pareek@leicester.ac.uk

## Abstract

### Background

Healthcare workers (HCWs), particularly those from ethnic minority groups, have been shown to be at disproportionately higher risk of infection with Severe Acute Respiratory Syndrome Coronavirus 2 (SARS-CoV-2) compared to the general population. However, there is insufficient evidence on how demographic and occupational factors influence infection risk among ethnic minority HCWs.

### Methods and findings

We conducted a cross-sectional analysis using data from the baseline questionnaire of the United Kingdom Research study into Ethnicity and Coronavirus Disease 2019 (COVID-19) Outcomes in Healthcare workers (UK-REACH) cohort study, administered between December 2020 and March 2021. We used logistic regression to examine associations of

---

**Data Availability Statement:** To protect the confidentiality of our participants, UK-REACH data

cannot be made freely available. To access data or samples produced by the UK-REACH study, the working group representative must first submit a request to the Core Management Group by contacting the UK-REACH Project Manager in the first instance (uk-reach@leicester.ac.uk). For ancillary studies outside of the core deliverables, the Steering Committee will make final decisions once they have been approved by the Core Management Group. Decisions on granting the access to data/materials will be made within eight weeks. Third party requests from outside the Project will require explicit approval of the Steering Committee once approved by the Core Management Group. Note that should there be significant numbers of requests to access data and/or samples then a separate Data Access Committee will be convened to appraise requests in the first instance.

**Funding:** UK-REACH is supported by a grant from the MRC-UK Research and Innovation (MR/V027549/1) and the Department of Health and Social Care through the National Institute for Health Research (NIHR) rapid response panel to tackle COVID-19. Core funding was also provided by NIHR Biomedical Research Centres. CAM is an NIHR Academic Clinical Fellow (ACF-2018-11-004). DP is supported by the NIHR. KW is funded through an NIHR Career Development Fellowship (CDF-2017-10-008). LBN is supported by an Academy of Medical Sciences Springboard Award (SBF005\1047). ALG was funded by internal fellowships at the University of Leicester from the Wellcome Trust Institutional Strategic Support Fund (204801/Z/16/Z) and the BHF Accelerator Award (AA/18/3/ 34220). MDT holds a Wellcome Trust Investigator Award (WT 202849/Z/ 16/Z) and an NIHR Senior Investigator Award. KK and LJG are supported by the National Institute for Health Research (NIHR) Applied Research Collaboration East Midlands (ARC EM). KK and MP are supported by the NIHR Leicester Biomedical Research Centre (BRC). MP is funded by a NIHR Development and Skills Enhancement Award (NIHR301192). This work is carried out with the support of BREATHE-The Health Data Research Hub for Respiratory Health [MC_PC_19004] in partnership with SAIL Databank. BREATHE is funded through the UK Research and Innovation Industrial Strategy Challenge Fund and delivered through Health Data Research UK. The funders had no role in study design, data collection and analysis, decision to publish, or preparation of the manuscript.

**Competing interests:** I have read the journal's policy and the authors of this manuscript have the

demographic, household, and occupational risk factors with SARS-CoV-2 infection (defined by polymerase chain reaction (PCR), serology, or suspected COVID-19) in a diverse group of HCWs. The primary exposure of interest was self-reported ethnicity.

Among 10,772 HCWs who worked during the first UK national lockdown in March 2020, the median age was 45 (interquartile range [IQR] 35 to 54), 75.1% were female and 29.6% were from ethnic minority groups. A total of 2,496 (23.2%) reported previous SARS-CoV-2 infection. The fully adjusted model contained the following dependent variables: demographic factors (age, sex, ethnicity, migration status, deprivation, religiosity), household factors (living with key workers, shared spaces in accommodation, number of people in household), health factors (presence/absence of diabetes or immunosuppression, smoking history, shielding status, SARS-CoV-2 vaccination status), the extent of social mixing outside of the household, and occupational factors (job role, the area in which a participant worked, use of public transport to work, exposure to confirmed suspected COVID-19 patients, personal protective equipment [PPE] access, aerosol generating procedure exposure, night shift pattern, and the UK region of workplace). After adjustment, demographic and household factors associated with increased odds of infection included younger age, living with other key workers, and higher religiosity. Important occupational risk factors associated with increased odds of infection included attending to a higher number of COVID-19 positive patients (aOR 2.59, 95% CI 2.11 to 3.18 for ≥21 patients per week versus none), working in a nursing or midwifery role (1.30, 1.11 to 1.53, compared to doctors), reporting a lack of access to PPE (1.29, 1.17 to 1.43), and working in an ambulance (2.00, 1.56 to 2.58) or hospital inpatient setting (1.55, 1.38 to 1.75). Those who worked in intensive care units were less likely to have been infected (0.76, 0.64 to 0.92) than those who did not. Black HCWs were more likely to have been infected than their White colleagues, an effect which attenuated after adjustment for other known risk factors. This study is limited by self-selection bias and the cross sectional nature of the study means we cannot infer the direction of causality.

## Conclusions

We identified key sociodemographic and occupational risk factors associated with SARS-CoV-2 infection among UK HCWs, and have determined factors that might contribute to a disproportionate odds of infection in HCWs from Black ethnic groups. These findings demonstrate the importance of social and occupational factors in driving ethnic disparities in COVID-19 outcomes, and should inform policies, including targeted vaccination strategies and risk assessments aimed at protecting HCWs in future waves of the COVID-19 pandemic.

## Trial registration

The study was prospectively registered at ISRCTN (reference number: ISRCTN11811602).

## Author summary

### Why was this study done?

- Previous research has shown that healthcare workers (HCWs), particularly those from ethnic minority groups, are at high risk of Coronavirus Disease 2019 (COVID-19).

following competing interests: MDT has had research collaborations with GlaxoSmithKline and Orion Pharma unrelated to the current work. KK is Director of the University of Leicester Centre for Black Minority Ethnic Health, Trustee of the South Asian Health Foundation and Chair of the Ethnicity Subgroup of the UK Government Scientific Advisory Group for Emergencies (SAGE). SC is Deputy Medical Director of the General Medical Council, UK Honorary Professor, University of Leicester. MP reports grants from Sanofi, grants and personal fees from Gilead Sciences and personal fees from QIAGEN, outside the submitted work. ALG Co-investigator on UK-REACH, which is supported by a grant from the MRC-UK Research and Innovation (MR/V027549/1) and the Department of Health and Social Care through the National Institute for Health Research (NIHR) rapid response panel to tackle COVID-19. Past funding by internal fellowships at the University of Leicester from the Wellcome Trust Institutional Strategic Support Fund (204801/Z/16/Z) and the BHF Accelerator Award (AA/18/3/ 34220). Member of Wellcome Trust Longitudinal Populations studies COVID-19 questionnaire steering group, to design national questionnaire to capture impact of COVID-19 pandemic on the population. KRA is a member of the National Institute for Health and Care Excellence (NICE) Diagnostics Advisory Committee, the NICE Decision and Technical Support Units, and is a National Institute for Health Research (NIHR) Senior Investigator Emeritus [NF-SI-0512-10159]. He has served as a paid consultant, providing unrelated methodological and strategic advice, to the pharmaceutical and life sciences industry generally, as well as to DHSC/ NICE, and has received unrelated research funding from Association of the British Pharmaceutical Industry (ABPI), European Federation of Pharmaceutical Industries & Associations (EFPIA), Pfizer, Sanofi and Swiss Precision Diagnostics/ Clearblue. He has also received course fees from ABPI and is a Partner and Director of Visible Analytics Limited, a health technology assessment consultancy company.

**Abbreviations:** aOR, adjusted odds ratio; AGP, aerosol generating procedure; CI, confidence interval; COVID-19, Coronavirus Disease 2019; HCW, healthcare worker; ICU, intensive care unit; IMD, index of multiple deprivation; IQR, interquartile range; OR, odds ratio; ONS, Office for National Statistics; PCR, polymerase chain reaction; PPE, personal protective equipment; SARS-CoV-2, Severe Acute Respiratory Syndrome Coronavirus 2; UK-REACH, United Kingdom Research study into Ethnicity and COVID-19 diagnosis and outcomes in Healthcare workers.

- This study aims to provide information about the reasons why these groups face a high risk of COVID-19.

## What did the researchers do and find?

- We conducted an electronic survey of over 12,000 United Kingdom HCWs (30% of whom were from ethnic minority groups) as part of the United Kingdom Research study into Ethnicity and COVID-19 diagnosis and outcomes in Healthcare workers (UK-REACH) study, in order to gather information about home and work factors that might be associated with COVID-19.

- Home factors associated with a higher risk of infection included younger age and living with other "key workers". Occupational factors associated with a higher risk of infection included attending to higher numbers of COVID-19 patients, working in a nursing or midwifery role, reporting a lack of access to appropriate personal protective equipment (PPE), and working in hospital inpatient and ambulance settings.

- HCWs from certain ethnic minority groups were at higher risk of COVID-19 than White HCWs. There are differences in home and occupational factors that affect COVID-19 risk between ethnic groups.

## What do these findings mean?

- We have identified key risk factors for COVID-19 in UK HCWs and have demonstrated that ethnic groups differ in home and work factors that affect COVID-19 risk. Therefore, these factors may explain the high risk of COVID-19 faced by ethnic minority HCWs.

- Our findings can be used to identify and protect "at risk" HCWs as the pandemic continues.

- As with all studies of this kind, caution is required when interpreting our findings as our results may be influenced by the differences between the group of HCWs who completed our survey compared to the healthcare workforce as a whole.

## Introduction

The first patients with Coronavirus Disease 2019 (COVID-19) in the United Kingdom (UK) were identified in late January 2020 [1]. Thousands of healthcare workers (HCWs) in the UK have since been infected with Severe Acute Respiratory Syndrome Coronavirus 2 (SARS-CoV-2) [2,3]. A report by Public Health England suggested that early in the pandemic, up to 73% of infections in HCWs were due to nosocomial transmission [2]. However, there remains insufficient evidence around key risk factors for infection in HCWs, and particularly what is driving reported ethnic disparities in infection risk. A recent study in the United States of America of over 24,000 HCWs found community exposures to be important in driving SARS-CoV-2 seropositivity but found no occupational predictors of infection [4], whereas a study of the workforce in 1 hospital in the UK found occupational factors to be important predictors of

SARS-CoV-2 seropositivity [3]. The specific underlying factors contributing to an increased risk of COVID-19 among HCWs from some ethnic minority groups, compared to White groups, are also poorly understood [5].

We sought to address these knowledge gaps using data from the national United Kingdom Research study into Ethnicity and COVID-19 diagnosis and outcomes in Healthcare workers (UK-REACH) longitudinal cohort study, which is among the largest UK HCW cohort studies and is unique in the richness of its dataset and the ethnic diversity of its participants. Specifically, we sought to determine risk factors for infection in UK HCWs and whether any disproportionate risks of infection in HCWs from ethnic minority groups might be explained by these risk factors.

## Methods

This study is reported as per the Strengthening the Reporting of Observational Studies in Epidemiology (STROBE) guideline (S1 Checklist) and the Checklist for Reporting Results of Internet E-Surveys (CHERRIES) (S2 Checklist) [6,7].

### Overview

UK-REACH is a programme of work aiming to determine the impact of the COVID-19 pandemic on UK HCWs, and establish whether, and to what degree, this differs according to ethnicity. This cross sectional analysis uses data from the baseline questionnaire of the prospective nationwide cohort study, administered between December 2020 and March 2021.

Details of the study design, sampling, and measures included in the baseline questionnaire can be found in the study protocol [8] and the data dictionary (https://www.uk-reach.org/data-dictionary).

### Study population

We recruited individuals aged 16 years or over, living in the UK and employed as HCWs or ancillary workers in a healthcare setting and/or registered with one of the following UK professional regulatory bodies: the General Medical Council, Nursing and Midwifery Council, General Dental Council, Health and Care Professions Council, General Optical Council, General Pharmaceutical Council, or the Pharmaceutical Society of Northern Ireland.

### Recruitment

We asked professional regulators to distribute emails to their registrants embedded with a hyperlink to the study website. The sample was supplemented by direct recruitment of participants through participating healthcare trusts, and advertising on social media and in newsletters. The study website contained information about the broad aim of the study (i.e., to investigate the impact of ethnicity on COVID-19 outcomes in HCWs). Entry into a prize draw to win 1 of 10 £250 vouchers was offered as an incentive for participation. Those interested could create a user profile, read the participant information sheet, and, if they were willing, sign an online consent form. After providing consent, participants were asked to complete the questionnaire.

### Questionnaire

The questionnaire was developed by the research team and tested for usability and technical functionality by the research team and by the UK-REACH Professional Expert Panel. Questionnaire items were not randomized. Branching logic was used to reduce number and

complexity of questions; therefore, the number of items on each page and the number of pages could vary significantly depending on answers and so we do not report this information. There was no "completeness check" at the end of the questionnaire. Respondents were able to change answers using a back button if needed. Duplicate records, which were created when a user did not save their questionnaire progress and returned to the questionnaire later leading to creation of a new record, were identified using a combination of study ID and a unique security token. In the case of duplicate entries, the more complete entry was kept and the other removed from the dataset.

## Outcome measures

Our primary outcome was SARS-CoV-2 infection, as determined by the self-reporting of either a positive polymerase chain reaction (PCR) assay for SARS-CoV-2 or a positive anti-SARS-CoV-2 serology assay. In addition, to ensure those that who acquired infection prior to widespread testing availability were not excluded, in those who had never been tested by PCR or serology, we included those individuals whose infection status was based on whether they, or another healthcare professional, suspected them of having had COVID-19 (see S1 Table for details).

## Covariates

Our primary exposure of interest was self-reported ethnicity, categorised using the UK's Office for National Statistics (ONS) 5- and 18-level ethnic group categories [9]. For the main analysis, ethnicity was categorised into 5 broad ethnic groups (White, Asian, Black, Mixed, and Other) to maximise the statistical power to test differences between groups. To ensure that we did not overlook important findings through collapsing ethnicity into broad groups, we also conducted additional analyses using 18 ethnicity categories.

   Other variables potentially associated with the outcome were selected a priori based on the existing literature and expert opinion. These comprised:

- Demographic characteristics (age, sex).

- Occupational factors (job role, area of work, number of confirmed/suspected COVID-19 patients seen per week with physical contact, sharing transport to work with those outside of the household, access to personal protective equipment [PPE], exposure to aerosol generating procedures [AGPs], hours worked per week, and night shift frequency).

- Household/residential/social factors (index of multiple deprivation [IMD, the official measure of relative deprivation for small areas of England, expressed as quintiles] [10], number of occupants in household, types of social contact [remote only, face-to-face with social distancing, or with physical contact], whether participants were cohabiting with another key worker [defined as someone expected to work during lockdown restrictions], and whether a participant's accommodation contained spaces that were shared with other households).

- Comorbidities (diabetes and immunosuppression) that might be associated with acquiring infection.

- SARS-CoV-2 vaccination status at the time of questionnaire response.

- Smoking status.

- UK region of workplace.

- Religiosity (i.e., how important a participant felt religion was in their daily life) and migration status were included to examine whether these might mediate any differences in

infection risk found between ethnic groups. We included religiosity rather than religion as it was felt that the relative importance of religion, and thus the inclination to attend religious gatherings/places of worship during the pandemic, was more important in terms of acquisition of SARS-CoV-2 infection than the specific religion of a participant.

A description of each variable and how it was derived from questionnaire responses can be found in S2 Table.

## Statistical analysis

We excluded those with missing data for the primary exposure (ethnicity) and outcome of interest (SARS-CoV-2 infection) from all analyses. Occupational variables used in the analysis reflect the participants' occupational circumstances during the weeks after implementation of the first national lockdown in the UK (which began on March 23, 2020). Therefore, in the main analysis, we excluded those not working during this time. We undertook an additional analysis examining demographic and home factors only in all participants. We kept the region of workplace variable in the model for the analyses of all participants as a proxy for the area in which the participant lives (to protect the confidentiality of our participants, researchers undertaking the analyses did not have access to residential postcodes).

We summarised categorical variables as frequency and percentage, and nonnormally distributed continuous variables as median (interquartile range [IQR]). We compared demographic, household, and occupational factors between ethnic groups using chi-square tests for categorical data and Kruskal–Wallis tests for continuous data.

We used univariable and multivariable logistic regression to determine unadjusted and adjusted associations of the variables described above with self-reported history of SARS-CoV-2 infection and report results as unadjusted and adjusted odds ratios (ORs and aORs) and 95% confidence intervals (95% CIs).

We reported frequency and percentage of observations with missing data for each variable of interest both overall and stratified by ethnicity.

Multiple imputation by chained equations was used to impute missing data in these logistic regression models. The imputation models included all variables used in the final analyses bar those being imputed, including the outcome measure. Rubin's rules were used to combine the parameter estimates and standard errors from 10 imputations into a single set of results [11]. Although indices of deprivation are available for UK countries outside England, it is recognised that these are not directly comparable with English IMD [12]. We therefore elected to code IMD as missing for those outside England and impute the missing information.

We undertook 3 sensitivity analyses. Firstly, an analysis was conducted including only HCWs who had been tested for evidence of SARS-CoV-2 infection by PCR or serology. Secondly, we undertook a complete case analysis. Thirdly, and finally, to account for the fact that antibodies to SARS-CoV-2 may be induced by vaccination, we recoded those determined to have been infected solely by a positive serology assay as uninfected if their antibody result date was both valid (as determined by its temporal association with questionnaire completion date) and later than their vaccination date.

To investigate the extent to which differences in infection risk by ethnic group could be explained by other related risk factors, we generated a base logistic regression model additionally adjusted for age and sex, and sequentially adjusted first for household/social/residential factors, second adding occupational risk factors, third adding health factors, fourth adding work region, and finally adding religiosity and migration status.

All analyses and multiple imputation were conducted using Stata 17 (StataCorp. 2021. Stata Statistical Software: Release 17. College Station, TX: StataCorp LLC).

### Changes to analysis in response to peer review

SARS-CoV-2 vaccination status was not initially included as a variable in our analyses. On peer review, it was suggested that we include this variable to take account of the impact of vaccination on transmission of infection, given that many HCW would have received at least 1 dose of vaccine during the study period. Originally, in our analysis of the effects of ethnicity on infection risk with sequential adjustment for groups of variables, religiosity, and migration status had been added to the model together. On peer review, it was suggested we add these separately.

### Ethical approval

The study was approved by the Health Research Authority (Brighton and Sussex Research Ethics Committee; ethics reference: 20/HRA/4718). All participants were given information on (i) the anticipated duration of the questionnaire; (ii) the data flows; (iii) the chief investigator; and (iv) the purpose of the study, after which they gave informed, written (electronic) consent. No personally identifiable information was collected in the questionnaire.

### Involvement and engagement

We worked closely with a Professional Expert Panel of HCWs from a range of ethnic backgrounds, healthcare occupations, and sexes, as well as with national and local organisations (see study protocol) [8].

## Results

### Cohort recruitment and formation of the analysis sample

The recruitment of the cohort has been described previously and details, including response rates, are shown in Fig 1 [8,13]. Briefly, 15,119 HCWs started the questionnaire, of whom 1,858 were excluded from the current analysis as they did not provide their ethnicity, and 720 were excluded due to a lack of outcome data. Therefore, 12,541 HCWs formed the analysis sample, 1,769 of whom were not working during lockdown and therefore were not included in analyses of occupational determinants of infection.

### Description of the analysed cohort

A description of the cohort is shown in Table 1. With reference to the cohort who were working during lockdown ($n$ = 10,772), the majority were female (75.1%) with a median age of 45 (IQR 35 to 54). Approximately 30% were from ethnic minority groups (19.1% Asian, 4.3% Black, 4.1% Mixed, 2.1% Other).

A description of the cohort who were working during lockdown, stratified by ethnicity, is shown in S3 Table. Almost all of the covariates differed by ethnicity. Age was different by ethnic group ($p < 0.001$), being lower in the Black and Asian cohorts compared to the White cohort (Black 43.5 [IQR 34.5 to 54], Asian 42 [IQR 33 to 51], and White 46 [IQR 36 to 55]). A greater proportion of Black HCWs lived in areas corresponding to lower IMD quintiles than White HCWs. Important differences were also demonstrated when examining religiosity by ethnic group ($p < 0.001$) with much greater proportions of Black and Asian HCWs describing their religion as being extremely important to their everyday lives compared to the White cohort (41.9% [Black], 19.5% [Asian] versus 5.9% [White]). Ethnic distribution was not equal across regions of the UK ($p < 0.001$) with a higher proportion of Black and Asian HCWs practicing in London (26.4% [Black], 21.3% [Asian] versus 11.8% [White]), and a lower proportion practicing in Scotland (2.5% [Black], 4.6% [Asian] versus 7.2% [White]) and South West

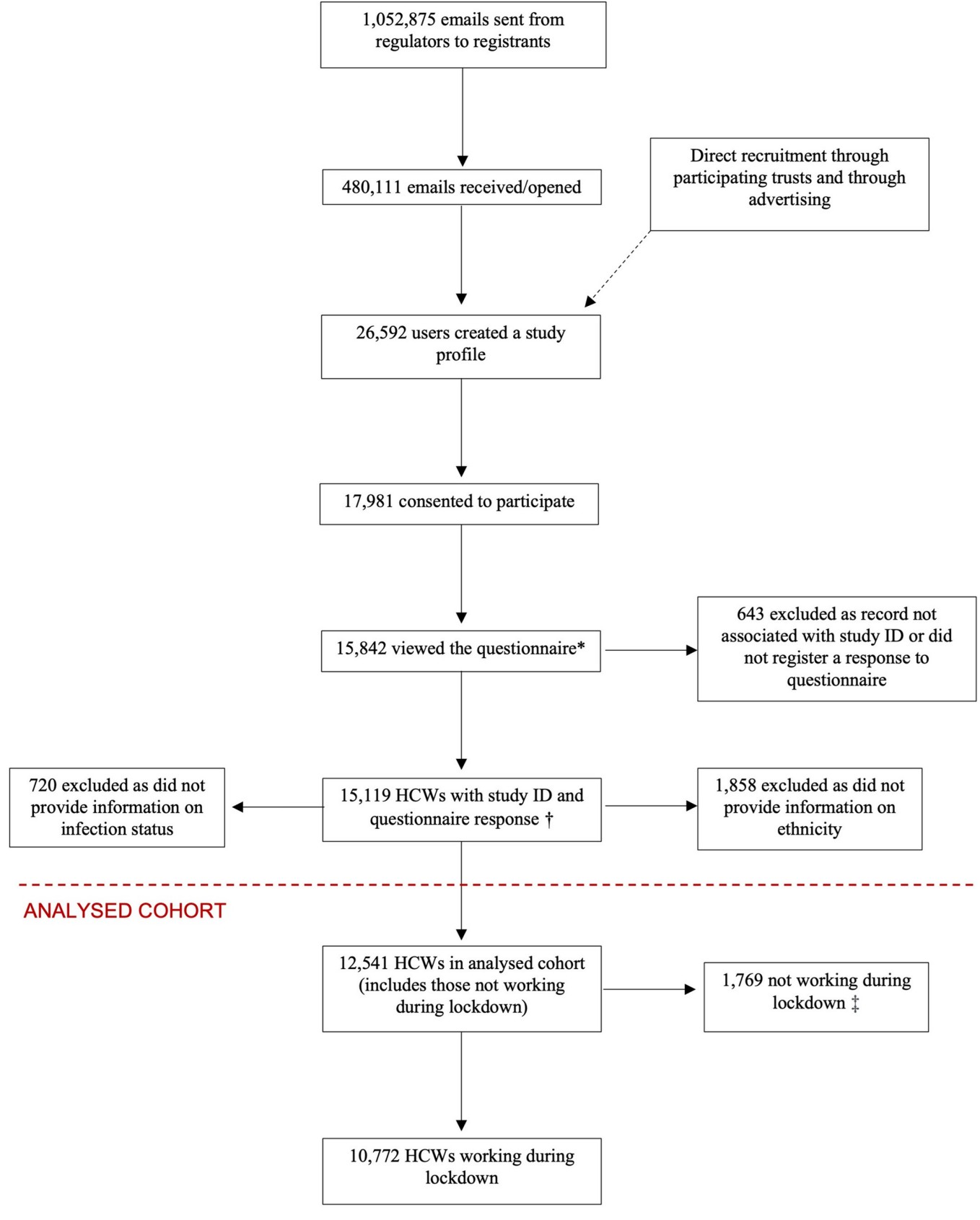

**Fig 1. HCW (those in professional healthcare roles or ancillary workers in a healthcare setting or registered with one of the 7 participating UK healthcare professional regulatory bodies—see Methods for a list of participating regulatory bodies).** * There were 15,997 views of the questionnaire, 155 duplicate records were removed leaving 15,842 unique HCW views. † Corresponds to an effective response rate of 57.1% of those who registered/created a profile on the study website (and 84.5% of those who consented, 1.4% of those who were sent an email, and 3.2% of those who opened the email). ‡ Not included in analyses involving occupational variables. A total of 12,402/15,199 HCWs answered the last question of the questionnaire corresponding to a completion rate of 81.6%. HCW, healthcare worker.

England (4.2% [Black], 5.2% [Asian] versus 10.1% [White]). A total of 62.0% of White HCWs did not have physical contact with COVID-19 patients, this compares to 49.1% in Asian and 49.6% in Black HCWs.

## Univariable analysis of risk factors for SARS-CoV-2 infection

**Demographic, household, and health risk factors.**   Overall, 2,496 (23.2%) of the 10,772 HCWs who worked during lockdown reported evidence of previous infection. Compared to the uninfected participants, the infected participants were younger, with a greater proportion of Black HCWs compared to White HCWs (OR 1.40, 95% CI 1.14 to 1.73, $p = 0.001$) (Table 2). Comparable patterns of association were seen when including HCWs who reported they were not working during lockdown (S4 Table).

**Occupational risk factors.**   The proportion of HCWs with a reported history of COVID-19 was higher among those who reported attending to higher numbers of patients with confirmed/suspected COVID-19 (with physical contact). A total of 16.4% of those that had no physical contact with COVID-19 patients were infected, compared to 40.0% of those that attended to ≥21 COVID-19 patients per week (Table 2).

## Multivariable analysis of risk factors for SARS-CoV-2 infection

**Demographic, household, and health risk factors.**   In the working cohort, older HCWs were less likely to be infected (aOR 0.92, 95% CI 0.88 to 0.97, $p = 0.001$ for each decade increase in age). HCWs that lived with other key workers, compared to those that did not, had a small increase in odds of infection (1.17, 1.06 to 1.30, $p = 0.002$). Those who described their religion as extremely important were more likely to report infection than those to whom religion was not important or were not religious (1.28, 1.08 to 1.51, $p = 0.004$). HCWs who had received at least 1 dose of SARS-CoV-2 vaccine were less likely to have been infected than HCW who were unvaccinated (0.62, 0.54 to 0.72, $p < 0.001$). Demographic and household risk factors were unchanged if those not working during lockdown were included (Table 3).

**Occupational risk factors.**   Compared to doctors, those working in nursing and midwifery roles were more likely to be infected (1.30, 1.11 to 1.53, $p = 0.001$). The odds of infection were higher for HCWs who attended to a higher number of confirmed COVID-19 patients (with physical contact), with those attending to ≥21 patients per week being 2 and a half times more likely to be infected compared to those who did not attend to any COVID-19 patients. Compared to those who either did not need PPE or reported access to appropriate PPE each time they needed it, those who reported not having access to appropriate PPE at all times were more likely to be infected (1.29, 1.17 to 1.43, $p < 0.001$). Working in ambulance (2.00, 1.56 to 2.58, $p < 0.001$) or hospital inpatient (1.55, 1.38 to 1.75, $p < 0.001$) settings were associated with higher odds of infection, while working in an intensive care unit (ICU) setting was associated with lower odds of infection (0.76, 0.64 to 0.92, $p = 0.003$), when compared to those not working in these settings. HCWs working in Scotland and South West England were at approximately half the odds of being infected compared to those working in the West Midlands (Table 3).

**Table 1. Description of the 2 analysed cohorts.**

| Variable | Excluding those not working during lockdown | All participants with nonmissing ethnicity and infection status |
|---|---|---|
| | Total $N$ = 10,772 | Total $N$ = 12,541 |
| **Demographic and household factors** | | |
| **Age**, med (IQR) | 45 (35–54) | 45 (34–54) |
| Missing | 54 (0.5%) | 68 (0.5%) |
| **Sex** | | |
| Male | 2,660 (24.7%) | 2,977 (23.7%) |
| Female | 8,089 (75.1%) | 9,535 (76.0%) |
| Missing | 23 (0.2%) | 29 (0.2%) |
| **Ethnicity** | | |
| White | 7,583 (70.4%) | 8,795 (70.1%) |
| Asian | 2,057 (19.1%) | 2,418 (19.3%) |
| Black | 462 (4.3%) | 535 (4.3%) |
| Mixed | 446 (4.1%) | 529 (4.2%) |
| Other | 224 (2.1%) | 264 (2.1%) |
| Missing | 0 (0.0%) | 0 (0.0%) |
| **Migration status** | | |
| Born in UK | 7,901 (73.5%) | 9,171 (73.1%) |
| Born abroad | 2,847 (26.5%) | 3,341 (26.6%) |
| Missing | 24 (0.2%) | 29 (0.2%) |
| **Religiosity** | | |
| Not religious or not important | 6,085 (56.5%) | 7,043 (56.2%) |
| Fairly important | 2,268 (21.1%) | 2,637 (21.0%) |
| Very important | 1,064 (9.9%) | 1,238 (9.9%) |
| Extremely important | 1,124 (10.4%) | 1,335 (10.7%) |
| Missing | 231 (2.1%) | 288 (2.3%) |
| **Household size,** med (IQR) | 2 (1–3) | 2 (1–3) |
| Missing | 8 (0.1%) | 11 (0.1%) |
| **Cohabitation** | | |
| Does not live with other key workers | 5,571 (51.7%) | 6,574 (52.4%) |
| Lives with other key workers | 5,145 (47.8%) | 5,892 (47.0%) |
| Missing | 56 (0.5%) | 75 (0.6%) |
| **Accommodation** | | |
| Does not have shared spaces | 8,807 (81.8%) | 10,248 (81.7%) |
| Has shared spaces | 1,905 (17.7%) | 2,221 (17.7%) |
| Missing | 60 (0.6%) | 72 (0.6%) |
| **IMD quintile** | | |
| 1 (most deprived) | 956 (8.9%) | 1,112 (8.9%) |
| 2 | 1,597 (14.8%) | 1,840 (14.7%) |
| 3 | 1,944 (18.1%) | 2,301 (18.4%) |
| 4 | 2,312 (21.5%) | 2,671 (21.3%) |
| 5 (least deprived) | 2,700 (25.1%) | 3,164 (25.2%) |
| Missing | 1,263 (11.7%) | 1,453 (11.6%) |
| **Social mixing** | | |
| None or all remote | 2,685 (25.0%) | 3,165 (25.2%) |
| Face to face (with SD) | 6,584 (61.4%) | 7,647 (61.0%) |
| Physical contact | 1,460 (13.6%) | 1,675 (13.4%) |

(*Continued*)

**Table 1.** (*Continued*)

| Variable | Excluding those not working during lockdown | All participants with nonmissing ethnicity and infection status |
|---|---|---|
| | Total *N* = 10,772 | Total *N* = 12,541 |
| Missing | 43 (0.4%) | 54 (0.4%) |
| **Comorbidities** | | |
| HCW without diabetes | 9,918 (92.1%) | 11,518 (91.8%) |
| HCW with diabetes | 400 (3.9%) | 479 (3.8%) |
| Missing | 454 (4.2%) | 544 (4.3%) |
| **Comorbidities** | | |
| Not immunosuppressed | 9,983 (92.7%) | 11,577 (92.3%) |
| Immunosuppressed | 335 (3.1%) | 420 (3.4%) |
| Missing | 454 (4.2%) | 544 (4.3%) |
| **Shielding status** | | |
| Not advised to shield | 10,324 (95.8%) | 11,936 (95.2%) |
| Advised to shield | 410 (3.8%) | 554 (4.4%) |
| Missing | 38 (0.4%) | 51 (0.4%) |
| **Smoking status** | | |
| Never/ex-smoker | 10,139 (94.1%) | 11,815 (94.2%) |
| Current smoker | 533 (5.0%) | 609 (4.9%) |
| Missing | 533 (5.0%) | 117 (0.9%) |
| **COVID-19 vaccination status (at the time of questionnaire response)** | | |
| Unvaccinated | 3,853 (35.8%) | 3,496 (27.9%) |
| Vaccinated | 4,939 (45.9%) | 5,510 (43.9%) |
| Missing | 1,980 (18.4%) | 3,535 (28.2%) |
| **Occupational factors** | | |
| **Occupation** | | |
| Doctor or medical support | 2,596 (24.1%) | - |
| Nurse, NA, or Midwife | 2,354 (21.9%) | - |
| Allied health professional* | 4,422 (41.1%) | - |
| Dental | 418 (3.9%) | - |
| Admin, estates, or other | 607 (5.6%) | - |
| Missing | 375 (3.5%) | - |
| **Method of commuting** | | |
| Alone or with members of household | 9,577 (88.9%) | - |
| With others outside household | 1,061 (9.9%) | - |
| Missing | 134 (1.2%) | - |
| **Number of SARS-CoV-2–positive patients attended to per week (with physical contact)** | | |
| None | 6,298 (58.5%) | - |
| 1–5 | 2,169 (20.1%) | - |
| 6–20 | 1,506 (14.0%) | - |
| ≥ 21 | 687 (6.4%) | - |
| Missing | 112 (1.0%) | - |
| **Access to appropriate PPE** | | |
| Not applicable or all/most of the time | 4,560 (42.3%) | - |
| Some of the time or less frequently | 6,182 (57.4%) | - |
| Missing | 30 (0.3%) | - |
| **AGP exposure** | | |

(*Continued*)

**Table 1.** (Continued)

| Variable | Excluding those not working during lockdown | All participants with nonmissing ethnicity and infection status |
|---|---|---|
| | Total *N* = 10,772 | Total *N* = 12,541 |
| Less than weekly exposure | 8,437 (78.3%) | - |
| At least weekly exposure | 2,296 (21.3%) | - |
| Missing | 39 (0.4%) | - |
| **Night shift pattern** | | |
| Never works nights | 7,543 (70.0%) | - |
| Works nights less than weekly | 1,796 (16.7%) | - |
| Works nights weekly or always | 1,317 (12.2%) | - |
| Missing | 116 (1.1%) | - |
| **Work areas** | | |
| Ambulance | 396 (3.7%) | - |
| Community clinical setting/primary care | 2,426 (22.5%) | - |
| Nonclinical community setting | 565 (5.3%) | - |
| Emergency department | 963 (8.9%) | - |
| Intensive care unit | 927 (8.6%) | - |
| Hospital inpatient | 2,759 (25.6%) | - |
| Hospital outpatient | 1,831 (17.0%) | - |
| Hospital nonclinical area or laboratory | 1,114 (10.3%) | - |
| Psychiatric hospital | 312 (2.9%) | - |
| Maternity | 344 (3.2%) | - |
| Nursing or care home | 242 (2.3%) | - |
| University | 220 (2.0%) | - |
| Home | 1,715 (15.9%) | - |
| Missing (range)† | 34–40 (0.3–0.4%) | - |
| **Work region** | | |
| London | 1,423 (13.2%) | - |
| South East England | 1,265 (11.7%) | - |
| South West England | 857 (8.0%) | - |
| East of England | 759 (7.1%) | - |
| East Midlands | 1,097 (10.2%) | - |
| West Midlands | 834 (7.7%) | - |
| North East England | 445 (4.1%) | - |
| North West England | 1,101 (10.2%) | - |
| Yorkshire and the Humber | 778 (7.2%) | - |
| Wales | 334 (3.1%) | - |
| Scotland | 626 (5.8%) | - |
| Northern Ireland | 130 (1.2%) | - |
| Missing | 1,123 (10.4%) | - |

*Also includes pharmacists, healthcare scientists, ambulance workers, and those in optical roles.

†When asked about work areas participants could select multiple answers, therefore the work areas variables are "dummy" variables comparing all those that did not select an area (reference) with all those that did. Given the similar amount of missing data for each of these dummy variables, we present a range of number of missing items and proportions.

All occupational factors (other than region of workplace) relate to work circumstances during the weeks following the first UK national lockdown on March 23, 2020.

AGP, aerosol generating procedure; COVID-19, Coronavirus Disease 2019; IMD, index of multiple deprivation; IQR, interquartile range; PPE, personal protective equipment; SARS-CoV-2, Severe Acute Respiratory Syndrome Coronavirus 2.

**Table 2. Description of the cohort working during lockdown stratified by SARS-CoV-2 infection status with unadjusted odds ratios for the association of covariates with infection.**

| Variable | Excluding those not working during lockdown | | | |
|---|---|---|---|---|
| | Not infected 8,276 (76.8%) | Infected 2,496 (23.2%) | Unadjusted OR (95% CI) | *P* value |
| | Demographic and household factors | | | |
| **Age**, med (IQR) | 46 (36–54) | 43 (32–52) | 0.83 (0.80–0.86) | <0.001 |
| **Sex** | | | | |
| Male | 2,023 (24.5%) | 637 (25.6%) | Ref | - |
| Female | 6,238 (75.5%) | 1,851 (74.4%) | 0.94 (0.85–1.04) | 0.25 |
| **Ethnicity** | | | | |
| White | 5,872 (71.0%) | 1,711 (68.6%) | Ref | - |
| Asian | 1,555 (18.8%) | 502 (20.1%) | 1.11 (0.99–1.24) | 0.08 |
| Black | 328 (4.0%) | 134 (5.4%) | 1.40 (1.14–1.73) | 0.001 |
| Mixed | 351 (4.2%) | 95 (3.8%) | 0.93 (0.74–1.17) | 0.54 |
| Other | 170 (2.1%) | 54 (2.2%) | 1.09 (0.80–1.49) | 0.59 |
| **Migration status** | | | | |
| Born in UK | 6,126 (74.2%) | 1,775 (71.2%) | Ref | - |
| Born abroad | 2,129 (25.8%) | 718 (28.8%) | 1.16 (1.05–1.29) | 0.003 |
| **Religiosity** | | | | |
| Not religious or not important | 4,733 (58.4%) | 1,352 (55.4%) | Ref | - |
| Fairly important | 1,745 (21.6%) | 523 (21.4%) | 1.05 (0.94–1.18) | 0.41 |
| Very important | 806 (10.0%) | 258 (10.6%) | 1.12 (0.96–1.31) | 0.15 |
| Extremely important | 815 (10.1%) | 309 (12.7%) | 1.32 (1.14–1.53) | <0.001 |
| **Household size,** med (IQR) | 2 (1–3) | 2 (1–3) | 1.05 (1.02–1.08) | 0.004 |
| **Cohabitation** | | | | |
| Does not live with other key workers | 4,401 (53.5%) | 1,170 (47.1%) | Ref | - |
| Lives with other key workers | 3,830 (46.5%) | 1,315 (52.9%) | 1.29 (1.18–1.41) | <0.001 |
| **Accommodation** | | | | |
| Does not have shared spaces | 6,808 (82.7%) | 1,999 (80.5%) | Ref | - |
| Has shared spaces | 1,420 (17.3%) | 485 (19.5%) | 1.17 (1.04–1.31) | 0.01 |
| **IMD quintile** | | | | |
| 1 (most deprived) | 694 (9.6%) | 262 (11.5%) | 1.22 (1.02–1.46) | 0.03 |
| 2 | 1,174 (16.3%) | 423 (18.5%) | 1.17 (1.01–1.36) | 0.04 |
| 3 | 1,491 (20.6%) | 453 (19.8%) | Ref | - |
| 4 | 1,777 (24.6%) | 535 (23.4%) | 0.99 (0.85–1.14) | 0.85 |
| 5 (least deprived) | 2,089 (28.9%) | 611 (26.8%) | 0.94 (0.82–1.08) | 0.41 |
| **Social mixing** | | | | |
| None or all remote | 2,005 (24.3%) | 680 (27.4%) | Ref | - |
| Face to face (with SD) | 5,155 (62.5%) | 1,429 (57.5%) | 0.82 (0.74–0.91) | <0.001 |
| Physical contact | 1,084 (13.2%) | 376 (15.1%) | 1.02 (0.89–1.19) | 0.74 |
| **Comorbidities** | | | | |
| Patients without diabetes | 7,622 (96.1%) | 2,296 (96.2%) | Ref | - |
| Patients with diabetes | 310 (3.9%) | 90 (3.8%) | 0.96 (0.75–1.22) | 0.73 |
| **Comorbidities** | | | | |
| Not immunosuppressed | 7,659 (96.6%) | 2,324 (97.4%) | Ref | - |
| Immunosuppressed | 273 (3.4%) | 62 (2.6%) | 0.75 (0.57–0.99) | 0.04 |
| **Shielding status** | | | | |
| Not advised to shield | 7,917 (96.0%) | 2,407 (96.9%) | Ref | - |
| Advised to shield | 332 (4.0%) | 78 (3.1%) | 0.77 (0.60–0.99) | 0.05 |
| **Smoking status** | | | | |

*(Continued)*

**Table 2.** (*Continued*)

| Variable | Excluding those not working during lockdown | | | |
| --- | --- | --- | --- | --- |
| | Not infected 8,276 (76.8%) | Infected 2,496 (23.2%) | Unadjusted OR (95% CI) | *P* value |
| Never/ex-smoker | 7,760 (94.6%) | 2,379 (96.3%) | Ref | - |
| Current smoker | 441 (5.4%) | 92 (3.7%) | 0.68 (0.54–0.85) | 0.001 |
| **COVID-19 vaccination status (at time of questionnaire response)** | | | | |
| Unvaccinated | 4,317 (52.6%) | 1,421 (57.7%) | Ref | - |
| Vaccinated | 3,896 (47.4%) | 1,043 (42.3%) | 0.77 (0.70–0.86) | <0.001 |
| | **Occupational factors** | | | |
| **Occupation** | | | | |
| Doctor or medical support | 1,966 (24.6%) | 630 (26.1%) | Ref | - |
| Nurse, NA, or midwife | 1,721 (21.6%) | 633 (26.3%) | 1.14 (1.00–1.29) | 0.05 |
| Allied health professional* | 3,443 (43.1%) | 979 (40.6%) | 0.88 (0.78–0.98) | 0.03 |
| Dental | 359 (4.5%) | 59 (2.5%) | 0.50 (0.38–0.67) | <0.001 |
| Admin, estates or other | 497 (6.2%) | 110 (4.6%) | 0.69 (0.55–0.86) | 0.001 |
| **Method of commuting** | | | | |
| Alone or with members of household | 7,425 (90.9%) | 2,152 (87.3%) | Ref | - |
| With others outside household | 748 (9.2%) | 313 (12.7%) | 1.44 (1.25–1.66) | <0.001 |
| **Number of SARS-CoV-2–positive patients attended to per week (with physical contact)** | | | | |
| None | 5,268 (64.3%) | 1,030 (41.7%) | Ref | - |
| 1–5 | 1,537 (18.8%) | 632 (25.6%) | 2.11 (1.88–2.36) | <0.001 |
| 6–20 | 971 (11.9%) | 535 (21.6%) | 2.83 (2.49–3.20) | <0.001 |
| ≥21 | 412 (5.0%) | 275 (11.1%) | 3.43 (2.90–4.05) | <0.001 |
| **Access to appropriate PPE** | | | | |
| Not applicable or all/most the time | 3,701 (44.9%) | 859 (34.5%) | Ref | - |
| Some of the time or less frequently | 4,548 (55.1%) | 1,634 (65.5%) | 1.55 (1.41–1.70) | <0.001 |
| **AGP exposure** | | | | |
| Less than weekly exposure | 6,613 (80.2%) | 1,824 (73.3%) | Ref | - |
| At least weekly exposure | 1,631 (19.8%) | 665 (26.7%) | 1.48 (1.33–1.64) | <0.001 |
| **Night shift pattern** | | | | |
| Never works nights | 6,013 (73.4%) | 1,530 (62.0%) | Ref | - |
| Works nights less than weekly | 1,230 (15.0%) | 566 (22.9%) | 1.81 (1.61–2.02) | <0.001 |
| Works nights weekly or always | 946 (11.6%) | 371 (15.0%) | 1.54 (1.35–1.76) | <0.001 |
| **Work areas†** | | | | |
| Ambulance | 241 (2.9%) | 155 (6.2%) | 2.21 (1.79–2.72) | <0.001 |
| Community clinical setting/primary care | 1,983 (24.0%) | 443 (17.8%) | 0.68 (0.61–0.76) | <0.001 |
| Nonclinical community setting | 465 (5.6%) | 100 (4.0%) | 0.70 (0.56–0.87) | 0.001 |
| Emergency department | 644 (7.8%) | 319 (12.8%) | 1.74 (1.51–2.00) | <0.001 |
| Intensive care unit | 677 (8.2%) | 250 (10.0%) | 1.25 (1.07–1.46) | 0.004 |
| Hospital inpatient | 1,851 (22.5%) | 908 (36.5%) | 1.99 (1.80–2.19) | <0.001 |
| Hospital outpatient | 1,404 (17.0%) | 427 (17.2%) | 1.01 (0.90–1.14) | 0.88 |
| Hospital nonclinical area or laboratory | 918 (11.1%) | 196 (7.9%) | 0.68 (0.58–0.80) | <0.001 |
| Psychiatric hospital | 224 (2.7%) | 88 (3.5%) | 1.31 (1.02–1.69) | 0.03 |
| Maternity | 278 (3.4%) | 66 (2.7%) | 0.78 (0.69–1.02) | 0.07 |
| Nursing or care home | 173 (2.1%) | 69 (2.8%) | 1.32 (1.00–1.75) | 0.05 |
| University | 177 (2.2%) | 43 (1.7%) | 0.80 (0.57–1.11) | 0.18 |
| Home | 1,444 (17.5%) | 271 (10.9%) | 0.57 (0.50–0.66) | <0.001 |
| **Work region** | | | | |
| West Midlands | 627 (8.4%) | 207 (9.4%) | Ref | - |

(*Continued*)

**Table 2.** (Continued)

| Variable | Excluding those not working during lockdown | | | |
| | Not infected 8,276 (76.8%) | Infected 2,496 (23.2%) | Unadjusted OR (95% CI) | P value |
|---|---|---|---|---|
| London | 1,037 (14.0%) | 386 (17.4%) | 1.12 (0.92–1.37) | 0.25 |
| South East England | 990 (13.3%) | 275 (12.4%) | 0.85 (0.69–1.04) | 0.12 |
| South West England (and Channel Islands) | 723 (9.7%) | 134 (6.1%) | 0.56 (0.45–0.72) | <0.001 |
| East of England | 599 (8.1%) | 160 (7.2%) | 0.82 (0.65–1.03) | 0.09 |
| East Midlands | 869 (11.7%) | 228 (10.3%) | 0.79 (0.64–0.98) | 0.03 |
| North East England | 342 (4.6%) | 103 (4.7%) | 0.90 (0.69–1.18) | 0.44 |
| North West England (and Isle of Man) | 773 (10.4%) | 328 (14.8%) | 1.28 (1.05–1.57) | 0.01 |
| Yorkshire and the Humber | 569 (7.7%) | 209 (9.4%) | 1.09 (0.88–1.37) | 0.41 |
| Wales | 246 (3.3%) | 88 (4.0%) | 1.09 (0.81–1.46) | 0.58 |
| Scotland | 549 (7.4%) | 77 (3.5%) | 0.44 (0.33–0.58) | <0.001 |
| Northern Ireland | 112 (1.5%) | 18 (0.8%) | 0.49 (0.29–0.82) | 0.007 |

Table 2 shows the cohort who worked during lockdown stratified by SARS-CoV-2 infection status, and unadjusted odds ratios for the association of covariates with SARS-CoV-2 infection.

* Also includes pharmacists, healthcare scientists, ambulance workers, and those in optical roles.

†When asked about work areas participants could select multiple answers, therefore, the work areas variables are "dummy" variables comparing all those that did not select an area (reference) with all those that did. Here, we only show the number and proportion of infected/noninfected participants who did select this area.

Percentages are computed column-wise other than the total of infected and noninfected HCWs, which are computed row-wise.

All occupational factors (other than region of workplace) relate to work circumstances during the weeks following the first UK national lockdown on March 23, 2020.

AGP, aerosol generating procedure; COVID-19, Coronavirus Disease 2019; HCW, healthcare worker; IMD, index of multiple deprivation; IQR, interquartile range; OR, odds ratio; PPE, personal protective equipment; Ref, reference category for categorical variables; SARS-CoV-2, Severe Acute Respiratory Syndrome Coronavirus 2; 95% CI, 95% confidence interval.

### Association of ethnicity with SARS-CoV-2 infection risk

In a model adjusted for age and sex, there was an increased risk of infection among Black HCWs compared to White HCWs (Fig 2). This association appeared to diminish as more variables were added to the model and, after adjustment for all covariates, differences in odds of infection between Black and White ethnic groups had attenuated.

### Sensitivity analyses

Results of (i) an analysis using an outcome of infection defined by either positive PCR or antibody and excluding those who had never been tested; (ii) an analysis of complete cases; and (iii) an analysis investigating the effect of vaccination-induced seropositivity on our results did not lead to any changes in our interpretation of the data (see S5–S7 Tables).

Univariable and multivariable logistic regression analyses using the more granular ethnicity categories are shown in S8 Table. In univariable analysis, those from Pakistani and Black African groups were more likely to be infected than their White British colleagues, but as with the main analysis, these effects were attenuated in the fully adjusted model.

## Discussion

In this analysis of over 12,000 UK HCWs, we found that nearly a quarter of participants reported having been infected with SARS-CoV-2 within the first year of the pandemic. The richness of the dataset and ethnic diversity of the cohort has allowed us to identify factors that may explain the disproportionate risks of infection between Black and White HCWs.

**Table 3. Multivariable analysis of factors associated with SARS-CoV-2 infection.**

| Variable | Adjusted for demographic, home and work factors during lockdown ($n$ = 10,772) | | Adjusted for demographic and home factors ($n$ = 12,541) | |
|---|---|---|---|---|
| | aOR (95% CI) | *P* value | aOR (95% CI) | *P* value |
| **Demographic and household factors** | | | | |
| **Age**[*] | 0.92 (0.88–0.97) | 0.001 | 0.85 (0.82–0.88) | <0.001 |
| **Sex** | | | | |
| Male | Ref | - | Ref | - |
| Female | 1.03 (0.91–1.16) | 0.66 | 0.91 (0.82–1.00) | 0.06 |
| **Ethnicity** | | | | |
| White | Ref | - | Ref | - |
| Asian | 0.87 (0.74–1.01) | 0.07 | 0.84 (0.73–0.95) | 0.007 |
| Black | 0.97 (0.76–1.24) | 0.82 | 0.93 (0.74–1.16) | 0.50 |
| Mixed | 0.83 (0.65–1.07) | 0.16 | 0.85 (0.68–1.05) | 0.14 |
| Other | 0.79 (0.55–1.11) | 0.18 | 0.82 (0.60–1.11) | 0.21 |
| **Migration status** | | | | |
| Born in UK | Ref | - | Ref | - |
| Born abroad | 1.09 (0.96–1.24) | 0.17 | 1.09 (0.97–1.22) | 0.13 |
| **Religiosity** | | | | |
| Not important or not religious | Ref | - | Ref | - |
| Fairly important | 1.08 (0.95–1.22) | 0.25 | 1.09 (0.97–1.21) | 0.16 |
| Very important | 1.05 (0.88–1.25) | 0.58 | 1.16 (1.00–1.35) | 0.06 |
| Extremely important | 1.28 (1.08–1.51) | 0.004 | 1.29 (1.11–1.50) | 0.001 |
| **IMD** | | | | |
| 1 (most deprived) | 1.02 (0.85–1.23) | 0.82 | 1.16 (0.98–1.37) | 0.08 |
| 2 | 1.08 (0.92–1.26) | 0.36 | 1.11 (0.96–1.30) | 0.16 |
| 3 | Ref | - | Ref | - |
| 4 | 1.01 (0.88–1.17) | 0.84 | 0.99 (0.86–1.14) | 0.91 |
| 5 (least deprived) | 1.05 (0.91–1.21) | 0.47 | 1.02 (0.89–1.16) | 0.82 |
| **Household size** | 1.02 (0.98–1.06) | 0.34 | 1.01 (0.97–1.04) | 0.67 |
| **Cohabitation** | | | | |
| Does not live with other key workers | Ref | - | Ref | - |
| Lives with other key workers | 1.17 (1.06–1.30) | 0.002 | 1.27 (1.16–1.39) | <0.001 |
| **Accommodation** | | | | |
| Does not have shared spaces | Ref | - | Ref | - |
| Has shared spaces | 0.93 (0.81–1.06) | 0.28 | 1.00 (0.89–1.13) | 0.95 |
| **Social mixing with others outside household** | | | | |
| None/remote only | Ref | - | Ref | - |
| Face to face with social distancing | 0.91 (0.81–1.01) | 0.09 | 0.90 (0.81–1.00) | 0.05 |
| With physical contact | 0.98 (0.84–1.15) | 0.85 | 1.04 (0.90–1.21) | 0.56 |
| **Comorbidities** | | | | |
| Diabetes | 1.11 (0.86–1.44) | 0.41 | 1.12 (0.89–1.40) | 0.32 |
| Immunosuppression | 1.00 (0.73–1.36) | 0.99 | 0.93 (0.70–1.23) | 0.60 |
| **Shielding status** | | | | |
| Not advised to shield | Ref | - | Ref | - |
| Advised to shield | 0.89 (0.66–1.18) | 0.41 | 0.81 (0.63–1.04) | 0.11 |
| **Smoking status** | | | | |
| Ex- or nonsmoker | Ref | - | Ref | - |
| Current smoker | 0.56 (0.44–0.72) | <0.001 | 0.63 (0.50–0.79) | <0.001 |
| **COVID-19 vaccination status (at time of questionnaire response)** | | | | |
| Unvaccinated | Ref | - | Ref | - |

*(Continued)*

**Table 3.** (Continued)

| Variable | Adjusted for demographic, home and work factors during lockdown ($n = 10,772$) | | Adjusted for demographic and home factors ($n = 12,541$) | |
|---|---|---|---|---|
| | aOR (95% CI) | P value | aOR (95% CI) | P value |
| Vaccinated | 0.62 (0.54–0.72) | <0.001 | 0.84 (0.75–0.94) | 0.003 |
| **Region of workplace** | | | | |
| West Midlands | Ref | - | Ref | - |
| London | 1.11 (0.90–1.38) | 0.33 | 1.13 (0.94–1.36) | 0.20 |
| South East England | 0.84 (0.68–1.05) | 0.12 | 0.88 (0.72–1.07) | 0.19 |
| South West England or Channel Islands | 0.58 (0.45–0.74) | <0.001 | 0.61 (0.49–0.77) | <0.001 |
| East of England | 0.76 (0.59–0.96) | 0.02 | 0.80 (0.64–0.99) | 0.04 |
| East Midlands | 0.88 (0.70–1.10) | 0.25 | 0.87 (0.71–1.07) | 0.18 |
| North East England | 0.87 (0.65–1.15) | 0.32 | 0.94 (0.73–1.22) | 0.65 |
| North West England or Isle of Man | 1.21 (0.98–1.49) | 0.08 | 1.25 (1.02–1.53) | 0.04 |
| Yorkshire and the Humber | 1.13 (0.89–1.43) | 0.33 | 1.10 (0.88–1.36) | 0.41 |
| Wales | 1.11 (0.82–1.50) | 0.51 | 1.13 (0.82–1.47) | 0.54 |
| Scotland | 0.44 (0.32–0.59) | <0.001 | 0.49 (0.38–0.63) | <0.001 |
| Northern Ireland | 0.47 (0.27–0.80) | 0.006 | 0.50 (0.30–0.81) | 0.005 |
| **Time between questionnaire rollout and questionnaire completion (per day)** | 1.00 (1.00–1.01) | 0.001 | 1.00 (1.00–1.00) | 0.87 |
| **Occupational factors** | | | | |
| **Occupation** | | | | |
| Doctor or medical support | Ref | - | - | - |
| Nurse, nursing associate, or Midwife | 1.30 (1.11–1.53) | 0.001 | - | - |
| Allied health professional† | 1.01 (0.87–1.16) | 0.94 | - | - |
| Dental | 0.70 (0.51–0.96) | 0.03 | - | - |
| Admin, estates, or other | 1.18 (0.91–1.53) | 0.21 | - | - |
| **Transport to work** | | | | |
| Alone or with members of household | Ref | - | - | - |
| With others outside household | 1.10 (0.93–1.29) | 0.27 | - | - |
| **Number of SARS-CoV-2–positive patients attended to per week (with physical contact)** | | | | |
| None | Ref | - | - | - |
| 1–5 | 1.70 (1.49–1.95) | <0.001 | - | - |
| 6–20 | 2.13 (1.82–2.50) | <0.001 | - | - |
| ≥21 | 2.59 (2.11–3.18) | <0.001 | - | - |
| **Access to appropriate PPE** | | | | |
| Not applicable or all/most of the time | Ref | - | - | - |
| Some of the time or less frequently | 1.29 (1.17–1.43) | <0.001 | - | - |
| **AGP exposure** | | | | |
| Less than weekly exposure | Ref | - | - | - |
| At least weekly exposure | 0.91 (0.80–1.04) | 0.17 | - | - |
| **Night shift pattern** | | | | |
| Never works nights | Ref | - | - | - |
| Works nights less than weekly | 1.10 (0.96–1.27) | 0.17 | - | - |
| Works nights weekly or always | 0.85 (0.72–1.00) | 0.05 | - | - |
| **Work areas** | | | | |
| Ambulance | 2.00 (1.56–2.58) | <0.001 | - | - |
| Community clinical setting/primary care | 0.93 (0.81–1.06) | 0.27 | - | - |
| Nonclinical community setting | 0.91 (0.72–1.15) | 0.45 | - | - |
| Emergency department | 1.10 (0.94–1.30) | 0.25 | - | - |
| Intensive care unit | 0.76 (0.64–0.92) | 0.003 | - | - |
| Hospital inpatient | 1.55 (1.38–1.75) | <0.001 | - | - |

(*Continued*)

**Table 3.** (Continued)

| Variable | Adjusted for demographic, home and work factors during lockdown (*n* = 10,772) | | Adjusted for demographic and home factors (*n* = 12,541) | |
| --- | --- | --- | --- | --- |
| | aOR (95% CI) | *P* value | aOR (95% CI) | *P* value |
| Hospital outpatient | 0.94 (0.82–1.07) | 0.36 | - | - |
| Hospital nonclinical area or laboratory | 0.87 (0.73–1.03) | 0.11 | - | - |
| Psychiatric hospital | 1.31 (1.00–1.71) | 0.05 | - | - |
| Maternity | 0.67 (0.50–0.89) | 0.006 | - | - |
| Nursing or care home | 1.34 (0.99–1.81) | 0.06 | - | - |
| University | 0.90 (0.63–1.29) | 0.57 | - | - |
| Home | 0.79 (0.68–0.92) | 0.002 | - | - |

Table 3 shows the results of 2 multivariable logistic regression analyses, 1 containing the whole analysed cohort, examining the association of demographic and household factors with infection, and the other containing the cohort working during lockdown, additionally adjusted for occupational factors.

*For each decade increase in age.

†Also includes pharmacists, healthcare scientists, ambulance workers, and those in optical roles.

Analyses adjusted for all other variables in the table (with the exception of the exclusion of occupational risk factors in the right-hand columns—as indicated by the lack of results in the relevant sections).

All occupational factors (other than region of workplace) relate to work circumstances during the weeks following the first UK national lockdown on March 23, 2020. When asked about work areas participants could select multiple answers, therefore the work areas variables are "dummy" variables comparing all those that did not select an area (reference) with all those that did. Region of workplace is included in the analysis of household and demographic factors as a proxy for the participants region of residence.

AGP, aerosol generating procedure; aOR, adjusted odds ratio; CI, confidence interval; COVID-19, Coronavirus Disease 2019; IMD, index of multiple deprivation; PPE, personal protective equipment; Ref, reference category for categorical variables; SARS-CoV-2, Severe Acute Respiratory Syndrome Coronavirus 2.

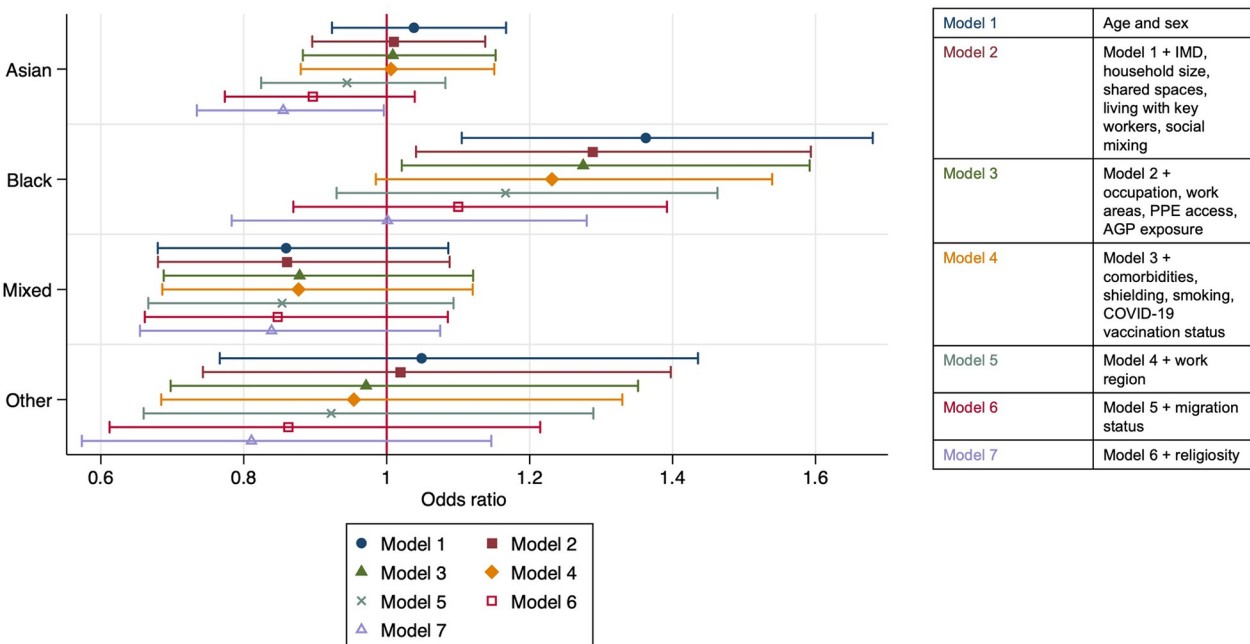

**Fig 2. Shows aORs (indicated by the central icon) and 95% CIs (indicated by the arms with caps at the lower and upper confidence interval) for the associations of the 5 broad ethnic groups (White ethnic group as reference) with SARS-CoV-2 infection and how these changed with sequential adjustment for groups of covariates.**

Additionally, we have identified important risk factors for SARS-CoV-2 infection in HCWs, including: working in nursing or midwifery roles, occupational exposure to increasing numbers of patients with COVID-19, lack of access to PPE, cohabiting with another key worker, and working in hospital inpatient or ambulance settings. Those working in particular UK regions (Scotland and South West England) had lower odds of infection than those working in the West Midlands, as did those working in ICU settings.

Our estimate of nearly a quarter of HCWs being infected with SARS-CoV-2 aligns with anti-SARS-CoV-2 seroprevalence estimates in UK healthcare settings, which have been reported to range from 10.8% to 44.0%, varying by UK region in which the study was conducted [3,14–16]. These estimates, including ours, are significantly higher than the estimated seroprevalence in England (prior to the vaccine rollout and after the first wave of the pandemic) which was estimated to be 6% [17], adding weight to the suggestion that UK HCWs are at higher risk of SARS-CoV-2 infection than the general population [17,18].

We demonstrate a strong association between the number of confirmed COVID-19 patients attended to by a HCW and their risk of infection. Previous studies have found conflicting evidence on this point. Caring for COVID-19 patients was found not to increase risk of infection in 2 large studies conducted in the USA [4,19], whereas in the UK, those working in patient-facing roles during the pandemic have been shown to be at higher risk of infection [3,16], and occupational exposure to patients and colleagues with COVID-19 has been shown to increase the risk of HCW infection in UK settings [20,21]. It should be noted that there are different PPE standards recommended by the 2 countries. In the USA, HCWs are advised to wear a "higher grade" of PPE (including an N95 respirator facemask even when not undertaking AGPs) than UK HCWs when attending to confirmed or suspected COVID-19 patients [22] and this may contribute to the differences in infection risk and the significant risk factors for infection for HCWs practising in the 2 countries.

Evidence for the importance of PPE in preventing HCW infection can be found in our analysis, with those who felt they did not have access to appropriate PPE at all times being more likely to have been infected than those who did. Furthermore, those working in ICU settings (where long sleeve gowns and respirator facemasks are recommended at all times) had lower odds of infection than those working elsewhere. These findings are in agreement with existing studies [3,14,18] and, together with the mounting evidence for aerosol transmission of SARS-CoV-2 [23,24], and the suggestion that coughing may generate more aerosols than activities designated as AGPs such as delivery of continuous positive airway pressure [25] support the claim that upgrading PPE standards for HCWs attending to COVID-19 patients (regardless of whether they are performing AGPs) may have a beneficial impact on the risk of HCW infection. It should be noted that the protective effect of working in an ICU setting against SARS-CoV-2 infection is likely to be multifactorial; in addition to improved PPE (when compared to other hospital and community settings), ICU staff may have a reduced number of patients to which they attend compared to staff on general wards (often having a 1:1 ratio of nursing staff to patients due to the intensity of the care requirements), closed circuit ventilators may prevent environmental contamination with SARS-CoV-2 and patients requiring ICU care may be at a later point in the disease course and may therefore be less infectious [26,27].

We found that a higher proportion of those from Black and Asian ethnic groups reported having been infected with SARS-CoV-2 compared to their White colleagues. This is commensurate with other studies conducted both in the USA and the UK [3,4,14,19,28]. Ethnicity is a complex construct; it has been defined as "the social group a person belongs to, and either identifies with or is identified with by others, as a result of a mix of cultural and other factors including language, diet, religion, ancestry, and physical features traditionally associated with race" [29]. Only by a deeper understanding of factors relating to disproportionate SARS-CoV-

2 infections in ethnic minority groups compared to White groups can we reduce hospitalisation, ICU admission, and death [30]. One strength of our study comes from the richness of our data, which allows us to determine the contribution that some of these interrelated factors may make to the higher risk of infection faced by HCWs from certain ethnic groups. In the fully adjusted model, there was no difference in the odds of infection between White and ethnic minority HCWs. This does not imply that there is not an increased risk of COVID-19 for ethnic minority HCWs, indeed it has been shown many times (and is apparent in this work in our univariable and age/sex adjusted analyses) that ethnic minority HCWs face an increased risk of SARS-CoV-2 infection compared to their White colleagues [3,14], rather it suggests that some of the covariates included in the fully adjusted model might drive the differences in the odds of infection by ethnicity.

Supporting evidence for this hypothesis can be found in our work as we found there to be an unequal ethnic distribution across other variables associated with increased odds of infection. For example, a far greater proportion of Black participants, compared to their White colleagues, worked in London and a far smaller proportion worked in Scotland, UK regions with among the highest and lowest infection rates respectively. Black HCWs reported far higher religiosity than White HCWs, which was shown to be associated with an increased likelihood of infection in the fully adjusted model. Black HCWs were also more likely to live in areas corresponding to the most deprived quintile and were more likely to have been born abroad than their White colleagues, both factors with a univariable association with higher infection risk. Importantly, our results indicate that sociodemographic and occupational differences between ethnic groups, such as those described above, are likely to be responsible for the increased probability of infection in Black HCWs compared to their White colleagues, as opposed to any innate biological characteristics.

It is reassuring, given the availability of SARS-CoV-2 vaccine to HCWs in the UK, that our study indicates that those who had accepted at least 1 vaccine against SARS-CoV-2 were at lower risk of COVID-19. However, when one considers that other studies have demonstrated clear differences in attitudes toward SARS-CoV-2 vaccination by ethnic group (with lower vaccine uptake and increased vaccine hesitancy demonstrated in some Black and Asian ethnic groups compared to White [13,31]) SARS-CoV-2 vaccination may represent another mediating factor in the relationship between ethnicity and COVID-19, which should be addressed to prevent even greater disparities in infection risk.

To our knowledge, we are the first to find that religiosity (one of the factors interrelated to a person's ethnicity) is associated with increased odds of SARS-CoV-2 infection. Religion has previously been associated with outcomes from COVID-19 at a population level, with analysis by the ONS showing those from particular religious groups (including Muslims and Hindus) in England and Wales are at higher risk of death from COVID-19 than Christians [32]. The mechanisms underlying our observation are unclear and warrant further investigation.

In our study, current smokers appear to have a lower risk of infection than former or nonsmokers. A recent meta-analysis found that current smokers (compared to never smokers) were at lower risk of testing positive for SARS-CoV-2 but at higher risk of hospitalisation and mortality [33] and a large seroprevalence study in the UK determined that there was a lower seroprevalence of anti-SARS-CoV-2 antibodies among current smokers than nonsmokers [17]. However, there are many factors that complicate the analysis of the effects of smoking on SARS-CoV-2, for example, smokers (because of a heightened awareness of their increased risk of respiratory disease) may be more likely to adhere to pandemic control measures [17], disruption of the mucosal epithelium in smokers may impact upon the sensitivity of PCR assays for acute infection and increase the odds of a negative test in the presence of disease [33]. Furthermore, some observational studies reporting on the effects of smoking on SARS-CoV-2

transmission have been hampered by collider bias, due to smokers being more likely to develop symptoms suggestive of COVID-19 for reasons other than SARS-CoV-2 infection and therefore more likely to test negative [34]. Regardless of the reasons underlying this observation, it is clear that the negative consequences of smoking far outweigh any potential protective effect against SARS-CoV-2 infection.

Risk of infection differed by UK region of workplace with HCWs in South West England, Scotland, and Northern Ireland being at lower risk than those working in the West Midlands. Compared to the West Midlands, these areas have a lower population density and a lower proportion of the population are from ethnic minority groups [35–38], factors associated with a decreased risk of SARS-CoV-2 transmission [39,40]. Additionally, government imposed restrictions aimed at slowing viral transmission differed between the UK nations and this may have influenced the lower infection risk seen among those working in Scotland and Northern Ireland [39].

We found ambulance workers to be at twice the risk of infection compared to those not working in this setting. To our knowledge, we are the first to demonstrate this effect. The reasons underlying this association require further investigation, although may relate to the front-line position of these HCWs and their exposure to the most critically unwell COVID-19 patients, with much of this exposure occurring in comparatively uncontrolled settings outside of hospital. In line with previous work from the UK, we also found nurses/midwives to be at higher risk of infection than those in medical roles [3].

Increasing age was associated with lower odds of infection. This effect may be due to the close correlation of age with occupational seniority. Senior HCWs spend a greater proportion of their time engaged in managerial and administrative responsibilities, and less time engaged in direct patient care, compared to their junior colleagues. This may lead to less occupational exposure to SARS-CoV-2 and therefore lower infection risk [3]. Additionally, older HCWs have been shown to report better access to PPE (which may also be related to reduced patient contact compared to junior staff [41]).

Our study has a number of limitations. There was potential for selection/responder bias, however comparison with the NHS workforce, while not an ideal reference population, indicates our sample is broadly representative, albeit with a lower proportion of ancillary staff (bias in the UK-REACH cohort study has been explored elsewhere [13]). As with any consented cohort study, there is the potential for self-selection bias. Reanalysis with selection weights to account for this bias was considered, and may form the basis of future work using UK-REACH data; however, given the difficulties with determining a reference population (which are due to our deliberately broad inclusion criteria), conducting and validating this analysis was felt to be beyond the scope of the current work. The cross-sectional nature of the study means we cannot infer the direction of causality, since results may be vulnerable to reverse causation and residual confounding. HCWs who thought they had been infected prior to widespread testing and subsequently tested negative for SARS-CoV-2 infection later in the pandemic would be coded as uninfected in our analysis. We may, therefore, have underestimated infection prevalence. Reassuringly, as noted above, the proportion of infected HCWs is in-line with estimates from other UK studies. PCR and serology status are self-reported, although given the implications of positive SARS-CoV-2 tests in a HCW population, we do not expect recall bias to have much effect on our outcome measure. In using multiple imputation to impute missing data, we assumed that data were "missing at random," while we have no reason to believe that data are "missing not at random" it is not possible to be definitive about whether this assumption holds for every variable in our imputation models, and therefore, it is possible that the use of multiple imputation may have introduced bias. However, the proportion of missing data for each variable of interest are small and results of a complete case analysis are not dissimilar to

results obtained from imputed datasets. In our wider analysis of demographic and home factors in the wider cohort, we used workplace region as a proxy measure for area of residence, we acknowledge that this will not be accurate for all participants due to inter-region travel from home to work. Collapsing "access to PPE" into a binary variable and combining "not applicable" with "all/most of the time" could have affected results. Future studies exploring the relationship between PPE and infection risk may wish to separate these categories.

In conclusion, we identified key sociodemographic and occupational factors associated with SARS-CoV-2 infection among UK HCWs in a large national cohort study. These findings are of urgent public health importance, especially in light of the emergence of a highly transmissible variant of SARS-CoV-2 (omicron), against which vaccination may be less effective [42]. The results should inform policies aimed at protecting HCWs in future pandemic waves through individualised risk assessments, proactive vaccination strategies (including the booster vaccines) to those at highest risk, and better communication around drivers of infection risk to safeguard the healthcare workforce. Critically, we demonstrate that Black HCWs in the UK are more likely to contract COVID-19 than their White colleagues. We have identified some factors interrelated to ethnicity that may underlie this association. Further work should focus on examining how these factors might mediate any disproportionate infection risk to inform interventions. This is particularly important given the increased prevalence of SARS-CoV-2 vaccine hesitancy in ethnic minority HCWs [13].

## Supporting information

**S1 Checklist. STROBE checklist.**
(DOC)

**S2 Checklist. CHERRIES checklist.**
(DOCX)

**S1 Table. Relative contribution of PCR, serology and suspected COVID-19 to overall number of infections in both cohorts.** COVID-19, Coronavirus Disease 2019; PCR, polymerase chain reaction.
(DOCX)

**S2 Table. Derivation of covariates from questionnaire data.**
(DOCX)

**S3 Table. Description of the cohort working during lockdown stratified by ethnicity together with tests of association between predictor variables and ethnicity.** IQR, interquartile range; med, median; NA, nursing associate; PPE, personal protective equipment.
(DOCX)

**S4 Table. Description of cohort, including those not working during lockdown, by infection status.** CI, confidence interval; OR, odds ratio; Ref, reference category for categorical variable.
(DOCX)

**S5 Table. Univariable and multivariable analysis of factors associated with SARS-CoV-2 infection as defined by positive PCR or serology and excluding those never tested.** aOR, adjusted odds ratio; PCR, polymerase chain reaction; PPE, personal protective equipment; Ref, reference category for categorical variables; SARS-CoV-2, Severe Acute Respiratory Syndrome Coronavirus 2.
(DOCX)

**S6 Table. Multivariable analysis of factors associated with SARS-CoV-2 infection in complete cases.** aOR, adjusted odds ratio; PPE, personal protective equipment; Ref, reference category for categorical variables; SARS-CoV-2, Severe Acute Respiratory Syndrome Coronavirus 2.
(DOCX)

**S7 Table. Effects of potential vaccination induced seropositivity on results.** aOR, adjusted odds ratio; PPE, personal protective equipment; Ref, reference category for categorical variables; SARS-CoV-2, Severe Acute Respiratory Syndrome Coronavirus 2.
(DOCX)

**S8 Table. Univariable and multivariable analysis of factors associated with SARS-CoV-2 infection using a more granular ethnicity variable in those who worked during lockdown.** aOR, adjusted odds ratio; PPE, personal protective equipment; Ref, reference category for categorical variables; SARS-CoV-2, Severe Acute Respiratory Syndrome Coronavirus 2.
(DOCX)

## Acknowledgments

We would like to thank all the participants who have taken part in this study when the NHS is under immense pressure. We wish to acknowledge the Professional Expert Panel group (Amir Burney, Association of Pakistani Physicians of Northern Europe; Tiffanie Harrison; London North West University Healthcare NHS Trust; Ahmed Hashim, Sudanese Doctors Association; Sandra Kazembe, University Hospitals Leicester NHS Trust; Susie M. Lagrata (Co-chair), Filipino Nurses Association, UK and University College London Hospitals NHS Foundation Trust; Satheesh Mathew, British Association of Physicians of Indian Origin; Juliette Mutuyimana, Kingston Hospitals NHS Trust; Padmasayee Papineni (Co-chair), London North West University Healthcare NHS Trust; Tatiana Monteiro, University Hospitals Leicester NHS Trust), the UK-REACH Stakeholder Group, the Study Steering Committee, Serco, as well as the following people and organisations for their support in setting up the study from the regulatory bodies: Kerrin Clapton and Andrew Ledgard (General Medical Council), Caroline Kenny (Nursing and Midwifery Council), David Teeman and Lisa Bainbridge (General Dental Council), My Phan and Jenny Clapham (General Pharmaceutical Council), Angharad Jones (General Optical Council), Mark Neale (Pharmaceutical Society of Northern Ireland), and the Health and Care Professions Council.

We would also like to acknowledge the following trusts and sites who recruited participants to the study: Affinity Care, Berkshire Healthcare NHS Trust, Birmingham and Solihull NHS Foundation Trust, Birmingham Community Healthcare NHS Foundation Trust, Black Country Community Healthcare NHS Foundation Trust, Bridgewater Community Healthcare NHS Trust, Central London Community Healthcare NHS Trust, Chesterfield Royal Hospital NHS Foundation Trust, County Durham and Darlington Foundation Trust, Derbyshire Healthcare NHS Foundation Trust, Lancashire Teaching Hospitals NHS Foundation Trust, Lewisham and Greenwich NHS Trust, London Ambulance NHS Trust, Northern Borders, Northumbria Healthcare NHS Foundation Trust, Nottinghamshire Healthcare NHS Foundation Trust, Royal Brompton and Harefield NHS Trust, Royal Free NHS Foundation Trust, Sheffield Teaching Hospitals NHS Foundation Trust, South Central Ambulance Service NHS Trust, South Tees NHS Foundation Trust, St George's University Hospital NHS Foundation Trust, Sussex Community NHS Foundation Trust, University Hospitals Coventry and Warwickshire NHS Trust, University Hospitals of Leicester NHS Trust, University Hospitals Southampton NHS Foundation Trust, Walsall Healthcare NHS Trust, and Yeovil District Hospital NHS Foundation Trust.

The UK-REACH Study Collaborative group: Manish Pareek (Chief investigator), Laura Gray (University of Leicester), Laura Nellums (University of Nottingham), Anna L Guyatt (University of Leicester), Catherine John (University of Leicester), I Chris McManus (University College London), Katherine Woolf (University College London), Ibrahim Akubakar (University College London), Amit Gupta (Oxford University Hospitals), Keith R Abrams (University of York), Martin D Tobin (University of Leicester), Louise Wain (University of Leicester), Sue Carr (University Hospital Leicester), Edward Dove (University of Edinburgh), Kamlesh Khunti (University of Leicester), David Ford (University of Swansea), and Robert Free (University of Leicester).

## Author Contributions

**Conceptualization:** Christopher A. Martin, Sue Carr, Amit Gupta, Anna L. Guyatt, Catherine John, I Chris McManus, Laura B. Nellums, Martin D. Tobin, Katherine Woolf, Kamlesh Khunti, Laura J. Gray, Manish Pareek.

**Data curation:** Christopher A. Martin, Luke Bryant, Manish Pareek.

**Formal analysis:** Christopher A. Martin, Daniel Pan, Carl Melbourne, Lucy Teece, I Chris McManus, Keith R. Abrams, Laura J. Gray, Manish Pareek.

**Funding acquisition:** Sue Carr, Amit Gupta, Anna L. Guyatt, Catherine John, I Chris McManus, Laura B. Nellums, Martin D. Tobin, Katherine Woolf, Kamlesh Khunti, Keith R. Abrams, Laura J. Gray, Manish Pareek.

**Investigation:** Christopher A. Martin, Carl Melbourne, Lucy Teece, Manish Pareek.

**Methodology:** Christopher A. Martin, Daniel Pan, Carl Melbourne, Lucy Teece, Rebecca F. Baggaley, Anna L. Guyatt, Katherine Woolf, Keith R. Abrams, Laura J. Gray, Manish Pareek.

**Project administration:** Carl Melbourne, Lucy Teece, Luke Bryant, Amit Gupta, Anna L. Guyatt, Catherine John, Laura B. Nellums, Martin D. Tobin, Katherine Woolf, Keith R. Abrams, Laura J. Gray, Manish Pareek.

**Resources:** Avinash Aujayeb, Luke Bryant, Bindu Gregary, Anna L. Guyatt, I Chris McManus, Rubina Reza, Sandra Simpson, Stephen Zingwe, Laura J. Gray, Manish Pareek.

**Software:** Luke Bryant.

**Supervision:** Lucy Teece, I Chris McManus, Katherine Woolf, Kamlesh Khunti, Keith R. Abrams, Laura J. Gray, Manish Pareek.

**Validation:** Christopher A. Martin, Manish Pareek.

**Visualization:** Manish Pareek.

**Writing – original draft:** Christopher A. Martin, Daniel Pan, Joshua Nazareth, Laura J. Gray, Manish Pareek.

**Writing – review & editing:** Christopher A. Martin, Daniel Pan, Carl Melbourne, Lucy Teece, Avinash Aujayeb, Rebecca F. Baggaley, Luke Bryant, Sue Carr, Bindu Gregary, Amit Gupta, Anna L. Guyatt, Catherine John, I Chris McManus, Joshua Nazareth, Laura B. Nellums, Rubina Reza, Sandra Simpson, Martin D. Tobin, Katherine Woolf, Stephen Zingwe, Kamlesh Khunti, Keith R. Abrams, Laura J. Gray, Manish Pareek.

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
