## [Editor Report · Decision Letter 0]

4 Feb 2022

Dear Dr Pareek, 

Thank you for submitting your manuscript entitled "Predictors of SARS-CoV-2 infection in a multi-ethnic cohort of United Kingdom healthcare workers : a prospective nationwide cohort study (UK-REACH)" for consideration by PLOS Medicine.

Your manuscript has now been evaluated by the PLOS Medicine editorial staff and I am writing to let you know that we would like to send your submission out for external peer review.

Please re-submit your manuscript within two working days, i.e. by Feb 08 2022 11:59PM.

Kind regards,

Callam Davidson

Associate Editor

PLOS Medicine

---

## [Decision Letter · Decision Letter 1]

15 Mar 2022

Dear Dr. Pareek,

Thank you very much for submitting your manuscript "Predictors of SARS-CoV-2 infection in a multi-ethnic cohort of United Kingdom healthcare workers : a prospective nationwide cohort study (UK-REACH)" (PMEDICINE-D-22-00390R1) for consideration at PLOS Medicine. 

[LINK]

In light of these reviews, I am afraid that we will not be able to accept the manuscript for publication in the journal in its current form, but we would like to consider a revised version that addresses the reviewers' and editors' comments. Obviously we cannot make any decision about publication until we have seen the revised manuscript and your response, and we plan to seek re-review by one or more of the reviewers. 

We hope to receive your revised manuscript by Apr 05 2022 11:59PM. Please email us (plosmedicine@plos.org) if you have any questions or concerns.

We look forward to receiving your revised manuscript. 

Sincerely,

Callam Davidson, 

Associate Editor

PLOS Medicine

plosmedicine.org

Please update your title to ‘Risk factors associated with SARS-CoV-2 infection in a multi-ethnic cohort of United Kingdom healthcare workers (UK-REACH): a cross-sectional analysis’, or similar.

Please structure your abstract using the PLOS Medicine headings (Background, Methods and Findings, Conclusions).

Abstract Methods and Findings:

* Please include the years during which the baseline questionnaire was administered.

* Please include that the primary exposure of interest was self-reported ethnicity.

* Please include the important dependent variables that are adjusted for in the analyses.

* Please report summary demographic information (e.g. sex, age, proportion from minority ethnic groups).

* Please remove the trial registration information from the abstract (this is OK to include in the Methods section).

The Data Availability Statement (DAS) requires revision. 

* If the data are owned by a third party but freely available upon request, please note this and state the owner of the data set and contact information for data requests (web or email address). Note that a study author cannot be the contact person for the data.

* If the data are not freely available, please describe briefly the ethical, legal, or contractual restriction that prevents you from sharing it. Please also include an appropriate contact (web or email address) for inquiries (again, this cannot be a study author).

Please include continuous line numbering throughout your manuscript to facilitate future review.

Please update citations throughout to be non-superscript, in square brackets, and preceding punctuation (e.g. [1]).

Please include a completed CHERRIES checklist in your Supplementary materials (https://www.equator-network.org/reporting-guidelines/improving-the-quality-of-web-surveys-the-checklist-for-reporting-results-of-internet-e-surveys-cherries/).

Please include whether informed consent was written or verbal.

Please remove the ‘Role of the funding source’ section.

Please ensure that the study is reported according to the STROBE guideline, and include the completed STROBE checklist as Supporting Information. Please add the following statement, or similar, to the Methods: "This study is reported as per the Strengthening the Reporting of Observational Studies in Epidemiology (STROBE) guideline (S1 Checklist)."

Did your study have a prospective protocol or analysis plan? Please state this (either way) early in the Methods section.

The UK-REACH protocol is cited as a preprint but I believe it is now published in BMJ Open – please update the citation accordingly. 

Given the substantial selection effects, selection weights could be considered in a sensitivity analysis. 

Supplementary information: "PLOS does not permit ""data not shown.” Please provide the data in accordance with the PLOS data policy. Please contact me if you have any additional questions.

When referring to religiosity in the Discussion, please add ‘to our knowledge’ to the claim of primacy.

Please remove the ‘Funding’, ‘Data Availability Statement’, and ‘Declaration of interest’ sections from the end of the main text and ensure all relevant information is captured in your responses to the Submission Form. 

References:

* Please use et al only after listing the first six authors of a study. 

* Journal name abbreviations should be those found in the National Center for Biotechnology Information (NCBI) databases.

* Please add [preprint] to references 33, 35, 37, and 38.

* Reference 34 is listed as ‘in press’, please confirm this is still accurate. 

Comments from the reviewers:

Reviewer #1: "Predictors of SARS-CoV-2 infection in a multi-ethnic cohort of United Kingdom healthcare workers : a prospective nationwide cohort study (UK-REACH)" reports factors associated with risk of infection, through cross-sectional logistic regression analysis based on questionnaire data on some 10,772 UK healthcare workers, collected between December 2020 and March 2021 over the Internet on the study website. A number of risk factors were found to be significantly correlated with SARS-CoV-2 infection from adjusted multivariate analysis, particularly attending to a higher number of COVID-19 positive patients.

Such broad-based research towards mitigating infection risk for HCW is commendable. However, a number of issues might be addressed:

1. A key concern would be vaccination status apparently not being included as a risk factor/variable in the analyses (e.g. not listed in Table 3). From public statistics, about 20 million individuals had obtained their first dose of a vaccine by the end of the period (March 2021), and a single dose has been reported to reduce risk of infection by around 60% (e.g. "Vaccine effectiveness of the first dose of ChAdOx1 nCoV-19 and BNT162b2 against SARS-CoV-2 infection in residents of long-term care facilities in England (VIVALDI): a prospective cohort study", Shrotri et al., The Lancet Infectious Diseases 21(11), 2021). 

This would appear a potentially significant factor (also for adjustment), especially as one would expect at-risk HCWs to be amongst the earliest to have vaccine doses administered. Moreover, there appears some references to vaccination dates (Statistical Analysis section) and analysis of vaccination induced seropositivity (Supplementary Information). As such, analysis including vaccination status as a variable would appear critical, or if not possible, this might be stated as a major limitation.

2. In the Predictor Variables section, "Comorbidities (diabetes and immunosuppression) that might be associated with acquiring infection (as opposed to risk of severe disease)" was stated as a criteria for variable selection. The criteria of "infection as opposed to severe disease" might be clarified further if possible. In particular, how would a variable that is associated with both infection and severe disease be considered? Are there notable examples of variables associated with severe disease, but not infection?

3. In the Statistical Analysis section, it is stated that workplace region is a proxy for region in which the participant lived. This assumption might be justified further if possible, i.e. is inter-region travel from home to workplace expected, especially if the workplace is a major city?

4. In the Statistical Analysis section, it is stated that multiple imputation was used to impute missing data in these logistic regression models. The imputation procedure and percentage of missing data for each variable might be indicated, possibly in the supplementary material.

5. In the Statistical Analysis section, it is stated that sensitivity analysis for vaccination-induced positive antibody tests was attempted, according to vaccination date. As first raised in Point 1, the coverage of known vaccinations might be reported.

Minor issues:

6. In the Introduction section, it is stated that "Thousands of healthcare workers (HCWs) in the UK have since been infected with Severe Acute Respiratory Syndrome Coronavirus-2 (SARS-CoV-2)". While true, a citation might be provided if possible.

7. Given that the administration of the questionnaire was done online, it might be clarified as to whether participants could be verified as actual HCW on the study website, since it appears that only an email address was required from [13] (i.e. are the emails required to be work emails, or was other verification implemented?)

8. In the Univariable Analysis section, it is stated that "The proportion of HCWs with a reported history of COVID-19 infection was proportionate to the number of patients with confirmed/suspected COVID-19 a HCW attended to (with physical contact)". The use of "proportionate" might be reconsidered in this context, unless there is evidence of an approximately constant ratio of infection risk with quantity of COVID-19 patient attended to. Also, "confirmed/suspected COVID-19" might be "confirmed/suspected COVID-19 patients" here.

Reviewer #2: The authors should be congratulated for this important piece of work which, using a large cohort of UK health care workers operating across multiple sectors of the NHS, confirms the findings of previous smaller studies and builds upon them by providing further granularity around the risk of SARS-COV-2 infection amongst minority ethnic groups. Given the increased COVID-19 morbidity and mortality observed amongst these groups, these data are a welcome and timely contribution to the literature.

The methodology and analyses appear appropriate and robust; the potential limitations of the study, including self-selection bias are noted in the discussion.

I have two minor comments that the authors may wish to consider further, but do not require specific revisions to be made.

1. The reduced risk associated with working within ITU has been observed in other studies. Many hypotheses have been proposed as to why this may be the case: improved PPE, reduced patient burden (1:1 nursing), closed circuit ventilators, reduced infectivity by the time patients have reached the point of ITU admission. I am unaware of any data supporting any data favouring any of these hypotheses and it is likely multifactorial. However, only some of these possibilities have been mentioned in the discussion.

2. The association with current smoking also consistently reproduces across multiple studies, is a curious observation, but is not elaborated upon further.

Reviewer #3: Thank you for this well-written and important paper. Strengths include the diversity of the study population, includling HCWs from different ethnicities, as well as thorough statistical analyses and a-priori defined risk factors to be included. Some of the main findings are to be expected, such as a close relationship with contact of the number of infected patients / lack of protective equipment and increased odds of infection among HCWs. 

Rather, the key strength of the paper lies in unravelling the contribution of determinants to obseved ethnic disparities among HCWs in COVID19 risk. Thus, I feel that the findings are of interest, yet struggle with the exact interpretation of the main findings of the paper. The authors want to elucidate key determinants/risk factors of increased infection risk among ethnic disparities, but as a reader I am uncertain whether the data truly reveal/explain these differences. In univariate analyses, some determinants are identified - but the associations attenuate in multivariate analyses. What is the take away for the audience? What I take from it now, is that a host of occupational, environmental and demographic determinants appear to relate to increased odds of COVID19 infection, but ethnicity does not seem to be one them in multivariate analyses (in fact, asians have a lower odds of infection compared to whites in multivariate analyses). If authors agree, I believe this should be mentioned more clearly throughout the paper. Please find some additional comments below:

Major comments:

1. Given the cross-sectional nature and etiological nature of this study, I would recommend to replace 'predictors' with determinants. Predictor to me implies a longitudinal relationship for a prognostic study, here it rather seems that the interest is of cross-sectional and etiological origin. 

"Specific risk factors contributing to an increased risk of COVID-19 among HCWs from some ethnic minority groups are also poorly understood." This is an important sentence of the introduction, but it is unclear to me what specific risk factors the authors refer to and why these are so important, and why would expect these to differ among black or asian ethnicities compared to for instance white HCWs?

2. It appears that the authors also have data available on vaccination status ('supplementary information'), I think it would be valuable to stratify main results based on vaccination status to see whether this could explain the observed differences across ethnicities, as they also mention in the discussion ("These findings have important public health implications given the increased risk of SARS-CoV-2 vaccine hesitancy in UK HCWs from Black ethnic groups.6"

3. I want to congratulate the authors with the thorough statistical analyses, in particular I like the stratification on self-reported/objectively collected COVID19 infections, as well as the granular analyses on 18-level ethnicity categories.

4. when reporting on main findings of the study, please mention effect estimates and corresponding CIs to text (i.e. "This association appeared to diminish as more variables were added to the model and, after adjustment for all predictors, differences in odds of infection between Black and White ethnic groups had attenuated.")

5. the association particularly appears to change when adding religliosity and migration status in model 6. Providing an extra model 7 and 8 with only migration or religliosity added on top of the others would be interesting to see what particularly drives the association.

6. I like Figure 2, it provides an excellent overview. Perhaps mention how CIs are displayed. Consider color coding that is friendly to colorblind readers. Perhaps add reference category of whites on top?

Minor comments:

-a key focus is on ethnic differences, thus I would present some additional information on baseline characteristics in the abstract. Please also provide response rate as a percentage of total (invited) source population

-please provide the main limitations of the study in the abstract

-adhere to STROBE guidelines and add checklist as supplement

-ii) sensitivity analysis of complete cases; -> in methods this analysis is mentioned as final analysis, in results it is mentioned as second. please align for consistency.

-are the authors able to correlate religiosity to vaccination hesitancy?, thereby also able to explain "the mechanisms underlying our observation are unclear and warrant further investigation."

-what variables could have been added to address residual confouding?, and how is reverse causation an issue in this study? "The cross-sectional nature of the study means we cannot infer the direction of causality, since results may be vulnerable to reverse causation and residual confounding."

-adding percentages relative to source population within figure 1 would be welcome

Thank you for having me as reviewer. 

Reviewer #4: This paper addressed predictors of SARS-CoV-2 infection in UK healthcare workers using cross-sectional data from a large cohort study into ethnicity and COVID-19 outcomes. The methods are straightforward but appropriate, and I believe results will be of broad interest to PLoS Medicine readers. The paper is clearly written and the work appears to have been carried out to a high standard. I have just a few minor suggestions for improvements:

Minor comments

1. This authors state that multiple imputation was used to impute missing data, but details are very scanty.  Box 2 of Sterne et al 2009 (doi: https://doi.org/10.1136/bmj.b2393 ) gives some suggestions for reporting methods and results of multiple imputation, and I would suggest the authors consider providing a bit more information in light of these recommendations.

2. Throughout the authors talk about "significant" differences, by which I presume they mean p<0.05. While I understand that this remains common practice in many medical journals, the dichotomising of results into significant and non-significant has been widely criticised in the statistical community (eg. via the consensus ASA statement doi.org/10.1080/00031305.2016.1154108 and elsewhere (eg. doi: 10.1136/bmj.322.7280.226 ). The manuscript wouldn't lose anything (and, in my opinion would gain) if reference to "significant" results were removed, and results interpretted in light of the confidence intervals and/or p-values, but without the unnecessary dichotomising. 

3. In the adjusted analysis the factor most strongly associated with reduced risk of SARS-CoV-2 infection is smoking. The authors are right to caution against causal interpretation of the results, but given the striking result I think this finding at least deserves a comment and placing in the context of other findings about the smoking and SARS-CoV-2 infection risk.

4. A couple of references that seem to me relevant for understanding risk of infection in healthcare workers that the authors might wish to consider in the discussion: https://doi.org/10.1371/journal.pmed.1003816 and https://www.nature.com/articles/s41467-022-28291-y (both of these highlight the risk of infection from nosocomially infected patients in hospitals) 

5. In the discussion on PPE on page 20 it would be helpful to give more detailed information on PPE recommendations at the time of the study in UK hospitals/ ICUs and in the USA where the other studies discussed were conducted. Additionally, it might be useful to point out that PPE recommendations for hospital staff in England changed on Jan 17th 2022 https://www.gov.uk/government/publications/wuhan-novel-coronavirus-infection-prevention-and-control/covid-19-guidance-for-maintaining-services-within-health-and-care-settings-infection-prevention-and-control-recommendations

[LINK]

---

## [Decision Letter · Decision Letter 2]

3 May 2022

Dear Dr. Pareek,

Thank you very much for re-submitting your manuscript "Risk factors associated with SARS-CoV-2 infection in a multi-ethnic cohort of United Kingdom healthcare workers (UK-REACH): a cross-sectional analysis" (PMEDICINE-D-22-00390R2) for review by PLOS Medicine.

I have discussed the paper with my colleagues and the academic editor and it was also seen again by two reviewers. I am pleased to say that provided the remaining editorial and production issues are dealt with we are planning to accept the paper for publication in the journal.

[LINK]

We look forward to receiving the revised manuscript by May 09 2022 11:59PM.   

Sincerely,

Callam Davidson, 

Associate Editor 

PLOS Medicine

plosmedicine.org

Requests from Editors:

Please define healthcare worker (HCW) in the legend for Figure 1.

Please ensure all Supporting Information items are cited in-text using the format outlined here: https://journals.plos.org/plosmedicine/s/supporting-information

Related to the above, please ensure you cite the CHERRIES checklist in your Supporting Information in a similar manner to how you have cited your STROBE checklist (lines 108-109). 

Comments from Reviewers:

Reviewer #1: We thank the authors for addressing our previous comments, in particular the reanalysis including vaccination status, which as perhaps might be expected was a fairly important factor. A couple of points might be considered:

1. While the lowered infection risk for Scotland and South West England has been discussed (Line 428 onwards), Northern Ireland appears to have a similarly reduced risk too, compared to the reference region (West Midlands). This might be discussed too if possible.

2. In Table 3, for "Access to appropriate PPE", it might be considered as to whether combining "Not applicable" and "all/most of the time" to be appropriate.

Reviewer #3: Thank you for your excellent response. All of my comments have been addressed appropiately. With best wishes

Silvan Licher

[LINK]

---

## [Decision Letter · Decision Letter 3]

9 May 2022

Dear Dr Pareek, 

On behalf of my colleagues and the Academic Editor, Dr Elvin Geng, I am pleased to inform you that we have agreed to publish your manuscript "Risk factors associated with SARS-CoV-2 infection in a multi-ethnic cohort of United Kingdom healthcare workers (UK-REACH): a cross-sectional analysis" (PMEDICINE-D-22-00390R3) in PLOS Medicine.

When making the formatting changes, please also make the update below:

* Pertaining to Reviewer 1, Comment 4 (on PPE), please briefly state the limitations of the existing analysis, to allow future work possibly based on it to be placed in context.

PRESS

Sincerely, 

Callam Davidson 

Associate Editor 

PLOS Medicine